# Directional Rank Reduction for Backdoor Defense

## Abstract

Recent studies have indicated the effectiveness of neuron pruning for backdoor defense. In this work, we explore the limitations of pruning-based defense through theoretical and empirical investigations. We argue that pruning-based defense necessitates the removal of neurons that affect normal performance when the effect of backdoor is entangled across normal neurons. To address this challenge, we introduce a novel and theoretically justified approach termed *Directional Rank Reduction* (DRR). This method aims to identify and eliminate the toxic direction - the direction maximizing the mean difference between benign and poisoned features, leading to a misclassification when triggered. Our approach, extended from standard neuron pruning, compresses features along arbitrary directions, providing a more flexible and effective defense mechanism against backdoor attacks. Extensive experimental results overwhelmingly show that our method outperforms others, averaging a threefold reduction in ASR while maintaining comparable ACC to the most recent robust baselines.

## 1 Introduction

Pre-trained models and public datasets are widely adopted in today's deep learning Goodfellow et al. (2016) applications. However, they may also bring security concerns. For instance, a malicious third party may provide pre-trained models with a predetermined response to specific patterns, also known as a *backdoor* trigger. An attacker can additionally inject a small proportion of malicious data into the public dataset, a process termed backdoor poisoning attacks (Gu et al., 2019; Li et al., 2020b), in order to mislead the trained model. This malicious data may take the form of a particular pattern patched into benign data with a changed label to the desired target. During training, the model will learn the association between the trigger and the specified target label, so that when the pattern is patched, the infected model will misclassify the input as the attack target, but otherwise act normally.

Studies have demonstrated that infected neural networks trained on poisoned datasets tend to generate backdoor neurons, which are neurons specialized for detecting backdoor triggers (Liu et al., 2018; Wu & Wang, 2021; Zheng et al., 2022b;a). On this basis, pruning-based defense approaches are proposed, which involve studying the characteristics of backdoor neurons that differentiate them from benign neurons and pruning accordingly.

Pruning can be regarded as an operation to compress feature along some dimensions to align benign and poisoned features in the feature space. From this perspective, the effect of pruning has not been well justified. We demonstrate in this paper that this is usually ineffective and suboptimal because the backdoor trigger effects are not always aligned with fixed dimensions of the feature space. An illustration of a failure case is shown in the left of Figure 1, where the backdoor trigger affect both dimensions and pruning either one of them cannot successfully eliminate the effect. In this study, we tackle the fundamental issue of feature alignment by focusing on minimizing the residual between benign and poisoned features. Through theoretical formulation, we ascertain the necessity to compress features along the direction of the mean difference between benign and poisoned features. This specific direction is referred to as the toxic direction.

Since the statistics information for pure benign data and pure poisoned data is inaccessible in practice, we propose to approximate the target direction through maximizing the third central moment with rigorous theoretical justification. After obtaining the target direction, we construct a projection matrix to project out the toxic direction to recover the model, as illustrated on the right side of Figure 1. We

Figure 1: Illustration of how DRR handle the failure case of neuron pruning.

refer to this as *Directional Rank Reduction* (DRR) - a reduction of the *rank* of the feature space. DRR compresses feature along arbitrary directions, which is an extension of the standard neuron pruning operation that can only compress along fixed directions.

The contributions of this paper are three-fold:

- *New insight:* We demonstrate the necessity of disentanglement of backdoor and normal features in pruning-based defense through theoretical analysis, showing that standard neuron pruning is a suboptimal operation in removing backdoor.
- *Novel framework:* We introduce a theoretically justified framework to identify and project out the "toxic direction" – the direction that maximizes the mean difference between benign features and poisoned features. This approach allows for compression along arbitrary directions, unlike standard neuron pruning methods, and offers a more robust defense against backdoor attacks.
- *Superior performance:* Extensive experimental results demonstrate the superiority of our method. On average, our approach substantially reduces the ASR by 3x with a comparable ACC with the recently strong baselines.

## 2 PRELIMINARIES

In this work, we consider a multi-class classification problem with $M$ classes. The training set $\mathcal{D} = \{(\boldsymbol{x}_i, y_i)\}_{i=1}^N$ consists of $N$ independently and identically distributed instances $\boldsymbol{x}_i \in \mathbb{R}^{C \times H \times W}$ and the corresponding labels $y_i \in \{1, 2, ..., M\}$ sampled from the domain $\mathcal{X} \times \mathcal{Y}$. Here, $c$, $h$, and $w$ denote the number of channels, the height, and the width of the images, respectively.

Let $F(x; \theta)$ be an $L$-layer neural network with $\theta$ its parameters. We may denote $F(x; \theta)$ as $F(x)$ or $F$ for simplicity. For each layer $1 \le l \le L$, we have $F^{(l)} = f^{(l)} \circ \phi \circ f^{(l-1)} \circ \phi \circ \cdots \circ \phi \circ f^{(1)}$, where $f^{(l)}$ is a linear function (e.g., fully connected layer) and $\phi$ is a nonlinear activation function applied element-wise. We denote the weight matrix of a fully connected layer by $\boldsymbol{W}^{(l)} \in \mathbb{R}^{C_{in}^{(l)} \times C_{out}^{(l)}}$, and the set of $l^{th}$ layer's features by $\boldsymbol{X}^{(l)}$. Then the linear function can be represented by: $\boldsymbol{X}^{(l+1)} = \boldsymbol{X}^{(l)} \boldsymbol{W}$. Note that we do not consider any bias terms for simplicity, while the conclusion can be easily generalized to the case in which the linear transformations contain bias terms.

## 3 ANALYZING PRUNING-BASED DEFENSE

In this section, we present theoretical and empirical analysis of existing pruning-based defense.

### 3.1 QUANTIFYING UTILITY

We propose a simple framework to theoretically analyze pruning methods by quantifying the utility of pruning operations. While clean accuracy (ACC) and attack success rate (ASR) are common choices for empirical evaluation, they require intensive evaluation on the test set and are inconvenient for analysis. To simplify the analysis, we use the *rank* of the feature space, which represent the expressiveness of the model, to measure the general performance of the model: pruning $n$ neurons reduces the rank of the feature space by $n$. To quantify the degree (*i.e.*, the influence) of backdoor,

we first define the residual matrix between clean features $\boldsymbol{X}^{(l)}$ and poisoned features $\hat{\boldsymbol{X}}^{(l)}$:

$$\boldsymbol{R}^{(l)} = \boldsymbol{X}^{(l)} - \hat{\boldsymbol{X}}^{(l)}.$$

Note that we assume the poisoned features are known *only* to assist analysis, which does not hold in practical defense and our algorithm does not use that. The residual matrix characterizes how each benign feature shifts after adding a trigger to the input. Each column of $\boldsymbol{R}^{(l)}$ is the feature shift of a specific neuron. If the model does not have a corresponding backdoor, the residual matrix should have nearly zero norm. Hence, in backdoor defense, it is reasonable to reduce the norm of the residual matrix, which explicitly aligns benign and poisoned features. We define the **utility** accordingly:

$$\mathcal{U}(\gamma_n) = \|\boldsymbol{R}^{(l)}\| - \|\gamma_n(\boldsymbol{R}^{(l)})\| = \|\boldsymbol{R}^{(l)} - \gamma_n(\boldsymbol{R}^{(l)})\|,$$

where $\|\cdot\|$ is a pending matrix norm and $\gamma_n(\cdot)$ is the processing function on feature matrix with rank budget $n$. For example, with $\gamma_n$ represent a pruning operation, $\gamma_n(\boldsymbol{X}) = \boldsymbol{X}\boldsymbol{M}_{\mathcal{I}}$ zeros $n$ columns in $\boldsymbol{X}$ with index set $\mathcal{I}$. Note that $\mathrm{rank}(\boldsymbol{R}^{(l)} - \gamma_n(\boldsymbol{R}^{(l)})) = n$ always holds. Combining the rank budget and the proposed utility function, the defender's goal can be written as maximizing the reduction of the matrix norm after processing the weight matrix within the given rank budget $n$.

In our analysis, we adopt $L_{1,1}$ norm, which is a special case of the $L_{p,q}$ norm:

**Definition 1.** $(L_{1,1})$ *Given a matrix $\boldsymbol{A}$ of size $m \times n$ with entries $a_{ij}$, the norm $\|\cdot\|_{1,1}$ is defined as:*

$$\|\boldsymbol{A}\|_{1,1} = \sum_{i=1}^{m}\sum_{j=1}^{n}|a_{ij}|$$

We adopt this matrix norm due to its simplicity and to get rid of the influence from second-order statistics (the covariance), which leads to the following property:

**Proposition 1.** *Given a matrix $\boldsymbol{A}$ of size $n \times m$, where each row is a size $m$ vector that is i.i.d. sample from a distribution A. When $m$ goes to infinity, the following property holds:*

$$\frac{1}{n}\|\boldsymbol{A}\|_{1,1} = \sum_{i=1}^{m}\mathbb{E}(A_i).$$

By adopting $L_{1,1}$ norm, the difference in the norm of residual matrices before and after pruning only depends on the columns to be pruned, the utility of the pruning operation becomes:

$$\mathcal{U}(\gamma_n) = \sum_{i \in \mathcal{I}}\|\boldsymbol{R}^{(l)}_{(:,i)}\|_1,$$

where $\boldsymbol{R}^{(l)}_{(:,i)}$ denotes the $i^{th}$ column of the residual matrix. Under this objective, the pruning-based defense has an optimal solution, *i.e.*, *to iteratively prune columns with the largest norm.* In theory, the optimal utility of pruning one neuron ($|\mathcal{I}| = 1$) is upper-bounded by the maximum column norm.

### 3.2 LIMITATIONS OF EXISTING PRUNING-BASED DEFENSE

The above formulation reveals the limitation of existing pruning-based defense due to its lack of flexibility, *i.e.*, *the pruning performance depends on how column norms are distributed in the residual matrix.* Ideally, when the column norms are sparsely distributed, *i.e.*, only a few columns have large norms and the others are close to zero, pruning the dominant columns could significantly reduce the residual norm. However, from an adversarial perspective, when all column norms are large and evenly distributed, the utility of pruning either neuron is limited and far from enough to reduce the residual norm to zero. This corresponds to the toy example raised in Figure 1, in which both two neurons have significant non-zero residuals.

Instead of pruning the preset axis as neuron pruning, it is natural to consider whether we can "prune" flexibly along non-axis directions. By proposition 1, when considering this problem from an expectation perspective, we aim at reducing:

$$\mathbb{E}(\boldsymbol{R}^{(l)}) = \mathbb{E}(\boldsymbol{X}^{(l)} - \hat{\boldsymbol{X}}^{(l)}) = \mathbb{E}(\boldsymbol{X}^{(l)}) - \mathbb{E}(\boldsymbol{X}^{(l)}),$$

where we view $\boldsymbol{R}^{(l)}$, $\boldsymbol{X}^{(l)}$ and $\hat{\boldsymbol{X}}^{(l)}$ as random variables that composite the rows of the corresponding matrix. It is obvious that the optimal solution will be to align the distribution of $\boldsymbol{X}^{(l)}$ and $\hat{\boldsymbol{X}}^{(l)}$ by directly pruning along the direction of the expectation difference. This is easy when we have access to $\boldsymbol{X}^{(l)}$ and $\hat{\boldsymbol{X}}^{(l)}$ respectively. However, as a defender, it is infeasible to access to the poisoned dataset.

# 4 OUR APPROACH

## 4.1 THEORETICAL FOUNDATION

In our setting, the defender has only access to the dataset of a mixture of benign and poisoned data with unknown fractions. Hence, our work is to provide an approximation on the expectation difference through samples from the mixture distribution. Specifically, our method is based on the following assumptions:

**Assumption 1.** (Gaussian Mixture Assumption) *Let $\boldsymbol{X}$ and $\hat{\boldsymbol{X}}$ be random vectors following multivariate Gaussian distributions with parameters $\boldsymbol{X} \sim \mathcal{N}(\boldsymbol{\mu}, \boldsymbol{\Sigma})$ and $\hat{\boldsymbol{X}} \sim \mathcal{N}(\hat{\boldsymbol{\mu}}, \hat{\boldsymbol{\Sigma}})$, respectively. Assume that both $\boldsymbol{\mu}, \hat{\boldsymbol{\mu}} \in \mathbb{R}^m$ and $\boldsymbol{\Sigma}, \hat{\boldsymbol{\Sigma}} \in \mathbb{R}^{m \times m}$.*

This assumption is commonly used in analyzing feature space of neural networks (Zheng et al., 2022b).

**Assumption 2.** (Separation Dominance Assumption) *The Separation Dominance Assumption is defined as follows:*
$$\|\boldsymbol{\mu}_1 - \boldsymbol{\mu}_2\|_2^2 = C \max(\lambda_{max}(\Sigma_1), \lambda_{max}(\Sigma_2)),$$
*where*
$$C \gg 1.$$

This assumption gives an quantifiable separation between the benign cluster and the poisoned cluster, which provides convenience for the following analysis.

**Theorem 1.** *Given a two-component Gaussian mixture model with the following probability density function:*

$$p(\boldsymbol{X}) = \pi_1 \mathcal{N}(\boldsymbol{X}|\boldsymbol{\mu}_1, \Sigma_1) + \pi_2 \mathcal{N}(\boldsymbol{X}|\boldsymbol{\mu}_2, \Sigma_2),$$

*where $\mathcal{N}(\boldsymbol{X}|\mu, \Sigma)$ represents the multivariate normal distribution with mean $\mu$ and covariance matrix $\Sigma$. Let $\boldsymbol{d} = \boldsymbol{\mu}_1 - \boldsymbol{\mu}_2$ denotes the direction of the mean difference and $\boldsymbol{v}$ denotes an arbitrary direction. Under assumption 2, a lower bound of the third central moment is given by:*

$$\mathbb{E}[(\boldsymbol{v}^T(\boldsymbol{X} - \boldsymbol{\mu}))^3] \geq \pi_1 \pi_2 (\boldsymbol{d}^T \boldsymbol{d}) \left(\boldsymbol{v}^T \left(3\Sigma_1 + 3\Sigma_2 + 2\boldsymbol{d}\boldsymbol{d}^T\right) \boldsymbol{v}\right),$$

*and maximizing the lower bound gives:*

$$\cos(\boldsymbol{v}, \boldsymbol{d}) \geq \sqrt{C^2 - 9}/C.$$

Proof can be found at Appendix A. This theorem states that, under the above assumption, maximizing the third central moment of a direction in the mixture distribution of benign and poisoned samples yields a direction $\boldsymbol{v}$ that is approximately close to the expectation difference of the two clusters $\boldsymbol{d}$, with a lower bound on $\cos(\boldsymbol{v}, \boldsymbol{d})$ depends on a constant $C$ in assumption 2. For example, if C=5, then $\cos(\boldsymbol{v}, \boldsymbol{d}) \geq 0.8$, which is fairly close to 1. See our experiments for C in more practical settings, which is much larger than 5.

## 4.2 DE-BACKDOOR PROCESS

Following the analysis above, the complete backdoor removal process is divided into two main steps: 1) optimizing the target direction vector, and 2) projecting out this direction.

**Optimization** This stage is dedicated to identifying a direction vector that maximizes the projected expectation difference between benign and poisoned clusters. As per the theorem referred in theorem 2, this is approximately the direction with the maximal third central moment of the mixed distribution. Given a poisoned dataset—comprising a significant amount of benign data and a minor portion of poisoned data—we first feed the mixed data into the model, obtaining features in the $l^{th}$ layer, denoted as $\{\boldsymbol{x}_i^{(l)}\}_{i=1}^{|\mathcal{D}|}$. For each layer, a direction vector $\boldsymbol{v}^{(l)}$ is initialized, and the sample third central moment along this direction is maximized:

$$\max_{\boldsymbol{v}} \quad \frac{1}{|\mathcal{D}|} \sum_{i=1}^{|\mathcal{D}|} (\boldsymbol{v}^{(l)T}(\boldsymbol{x}_i^{(l)} - \frac{1}{|\mathcal{D}|} \sum_{j=1}^{|\mathcal{D}|} \boldsymbol{x}_j^{(l)}))^3$$
$$\text{s.t.} \quad \|\boldsymbol{v}^{(l)}\|_2 = 1,$$

where $\mathcal{D}$ represents the poisoned dataset, and $\boldsymbol{x}_i$ are samples from this dataset. It is possible, however, that the separation might not be significant in the current layers. To address this, a threshold hyperparameter $\tau$ for the objective value is introduced to evaluate whether the optimized direction is valid. Only directions with a sufficiently large third central moment are carried forward to the next step.

**Projection** Once we get an direction vector, we construct a projection matrix with it by:

$$\boldsymbol{P}^{(l)} = \boldsymbol{I}^{(l)} - \boldsymbol{v}^{(l)T}(\boldsymbol{v}^{(l)}\boldsymbol{v}^{(l)T})\boldsymbol{v}^{(l)},$$

where $\boldsymbol{I}^{(l)}$ is an identity matrix in $l^{th}$ layer. The term $\boldsymbol{v}^{(l)T}(\boldsymbol{v}^{(l)}\boldsymbol{v}^{(l)T})\boldsymbol{v}^{(l)}$ calculates the component of a vector along the direction $\boldsymbol{v}^{(l)}$, and by subtracting this from the identity matrix, the resultant projection matrix $\boldsymbol{P}$ is formed. When a vector $\boldsymbol{x}^{(l)}$ is multiplied by this projection matrix $\boldsymbol{P}$, the component of x in the direction of $\boldsymbol{v}^{(l)}$ is removed, resulting in a vector that is orthogonal to $\boldsymbol{v}^{(l)}$. For each layer, we calculates the projection matrix Hence, for each layer, we calculates the projection matrix and modifies the weight matrix as follows:

$$\hat{\boldsymbol{W}}^{(l)} = \boldsymbol{P}^{(l)}\boldsymbol{W}^{(l)}.$$

**Extension** It is not guaranteed that a single optimized direction can successfully remove a backdoor, perhaps because the assumptions do not hold strictly or because the residual matrix (the backdoor trigger effect) has a rank greater than 1. Therefore, we can extend our approach to cover more general cases by learning a series of orthogonal directions $\boldsymbol{v}_1, \boldsymbol{v}_2, \ldots, \boldsymbol{v}_n$ and projecting out all of them. Specifically, after finding these directions, we construct a matrix with these direction vectors as columns:

$$\boldsymbol{V}^{(l)} = [\boldsymbol{v}_1^{(l)T}, \boldsymbol{v}_2^{(l)T}, \ldots, \boldsymbol{v}_n^{(l)T}],$$

and construct a projection matrix followed by:

$$\boldsymbol{P}^{(l)} = \boldsymbol{I}^{(l)} - \boldsymbol{V}^{(l)T}(\boldsymbol{V}^{(l)}\boldsymbol{V}^{(l)T})\boldsymbol{V}^{(l)}.$$

This completes our approach.

Note that there is an orthogonal constraint in the optimization problem, and can be solved by manifold optimization on Stiefel manifold (Absil et al., 2009). Manifold optimization is a generalization of classical optimization methods to handle optimization problems on manifolds, which are spaces with a non-Euclidean structure. Stiefel manifold is a subset of the Euclidean space that consists of all orthonormal matrices, which can be used to find optimal solutions to problems with orthogonal constraints. The whole algorithm is illustrated in algorithm 1.

## 5 EXPERIMENTS

### 5.1 EXPERIMENTAL SETUP

**Datasets.** Following existing work (Tran et al., 2018; Hayase & Kong, 2020; Wu & Wang, 2021), we adopt CIFAR-10 (Krizhevsky et al., 2009) for evaluating our approach. CIFAR-10 consists of 60,000 $32 \times 32$ RGB images that are divided into 10 classes (50,000 for training and 10,000 for testing).

**Models.** We use various models for evaluation: ResNet-18 (He et al., 2016), WideResNet28-1 (Zagoruyko & Komodakis, 2016) and DenseNet-121 (Huang et al., 2017). They are trained for 150 epochs on CIFAR-10 with SGD optimizer. The initial learning rate is set to 0.1 and the momentum is set to 0.9. We adopt the cosine learning rate scheduler to adjust the learning rate. The batch size is set to 128 by default.

---

**Algorithm 1** DRR: Directional rank reduction

---

**Input:** $L$-layer neural network $F^{(L)}$ with a set of weight matrix $\{\boldsymbol{W}^{(l)} : l = 1, 2, \ldots, L\}$, number
of direction vector $k$, threshold hyperparameter $\tau$, poisoned dataset $\mathcal{D}$, criterion $\mathcal{Q}$ (we adopt third
central moment througout our paper).
**for** $l = 1$ to $L$ **do**
    Get the weight matrix $\boldsymbol{W}^{(l)} \in \mathbb{R}^{C_{in}^{(l)} \times C_{out}^{(l)}}$
    Define the Stiefel manifold $St(C_{in}^{(l)} \times C_{out}^{(l)}) = \{\boldsymbol{M} \in \mathbb{R}^{C_{in}^{(l)} \times C_{out}^{(l)}} : \boldsymbol{M}^T \boldsymbol{M} = \boldsymbol{I}_{C_{in}^{(l)} \times C_{out}^{(l)}}\}$
    Initialize an orthogonal matrix $\boldsymbol{H}_{(0)} = \boldsymbol{I}$
    Get a set of features $\boldsymbol{X}^{(l)} = F^{(l)}(\boldsymbol{X})$ with $\boldsymbol{X} \sim \mathcal{D}$
    Get pruning indices $\mathcal{I} = \arg\max \mathcal{Q}((\boldsymbol{X}^{(l)} \boldsymbol{H}_t)_{(:,i)})$
    **for** $t = 1$ to $T$ **do**
        Compute loss $\mathcal{L} = -\sum_{i \in \mathcal{I}} \mathcal{Q}((\boldsymbol{X}^{(l)} \boldsymbol{H}_t)_{(:,i)})$ and obtain gradient on tangent space $\nabla \mathcal{L}|_{\boldsymbol{H}_{(t)}}$
        Update with retraction $\boldsymbol{H} = \mathrm{R}_{\mathrm{H}_{(t)}}(-\alpha \nabla \mathcal{L}|_{\boldsymbol{H}_{(t)}})$
    **end for**
    Let $\boldsymbol{V}^{(l)} = [\boldsymbol{v}_1^{(l)T}, \boldsymbol{v}_2^{(l)T}, \ldots, \boldsymbol{v}_n^{(l)T}]$ where $\boldsymbol{v}_j^{(l)} \in \{\boldsymbol{v}_i : \mathcal{Q}(\boldsymbol{X}_{\boldsymbol{H}_{(:,i)}}^{(l)}) > \tau\}$
    Construct a projection matrix: $\boldsymbol{P}^{(l)} = \boldsymbol{I}^{(l)} - \boldsymbol{V}^{(l)T}(\boldsymbol{V}^{(l)} \boldsymbol{V}^{(l)T})\boldsymbol{V}^{(l)}$
    Update the weight matrix: $\hat{\boldsymbol{W}}^{(l)} = \boldsymbol{P}^{(l)} \boldsymbol{W}^{(l)}$
**end for**
**return** $\{\hat{\boldsymbol{W}}^{(l)} : l = 1, 2, \ldots, L\}$

---

Table 1: Comparison of the proposed DRR with three state-of-the-art backdoor defenses on CIFAR-10. The best results are denoted with bold font and the second best with an underline.

| Model | Attack | Backdoored ACC | Backdoored ASR | FP ACC | FP ASR | ANP ACC | ANP ASR | CLP ACC | CLP ASR | EP ACC | EP ASR | DRR ACC | DRR ASR |
|---|---|---|---|---|---|---|---|---|---|---|---|---|---|
| ResNet18 | BadNets | 93.06 | 99.96 | 85.92 | 2.69 | 89.79 | 8.71 | 91.29 | **2.42** | 91.83 | 9.99 | **92.01** | 3.18 |
| | BadNets (A2A) | 92.22 | 91.74 | 87.76 | 18.09 | 90.50 | 16.31 | 92.05 | **1.45** | **93.69** | 4.97 | 92.42 | 1.93 |
| | Blended | 94.40 | 100.00 | 84.25 | 6.50 | 92.64 | 4.40 | 88.41 | **0.38** | 93.11 | 10.33 | **94.40** | 2.31 |
| | CLA | 94.29 | 99.26 | 88.34 | 1.14 | 90.09 | 17.36 | 88.94 | 16.66 | **93.85** | 2.23 | 93.40 | **0.98** |
| | Average | 93.49 | 97.74 | 86.57 | 7.11 | 90.76 | 11.70 | 90.17 | 4.98 | **93.12** | 6.88 | 93.06 | 2.10 |
| WRN-28-1 | BadNets | 92.34 | 100.00 | 83.61 | 2.00 | 80.85 | 5.77 | 89.93 | **1.23** | **91.19** | 19.81 | 89.56 | 1.69 |
| | BadNets (A2A) | 91.98 | 91.82 | 87.76 | 18.09 | 88.76 | 5.75 | 89.87 | 2.09 | 88.84 | 2.80 | **91.94** | **1.29** |
| | Blended | 92.24 | 100.00 | 87.88 | 6.31 | 85.78 | 5.98 | 90.65 | 100.00 | 88.41 | 45.91 | **91.50** | **2.74** |
| | CLA | 93.46 | 99.19 | 87.84 | 21.21 | 90.18 | 3.69 | 87.80 | 16.61 | 91.02 | **2.76** | **91.41** | 6.19 |
| | Average | 93.28 | 97.80 | 86.77 | 11.90 | 86.39 | 5.30 | 89.56 | 29.98 | 90.59 | 17.96 | **91.10** | 2.98 |
| Average | | 93.00 | 97.75 | 86.67 | 9.50 | 88.57 | 8.50 | 89.88 | 17.61 | 91.49 | 12.35 | **92.08** | **2.54** |

**Attacks.** We evaluate DRR mainly against four typical attack strategies, including *BadNet* (Gu et al., 2019), *Blended backdoor attack* (Blended) (Chen et al., 2017), and *Clean label attack* (CLA) (Turner et al., 2019). For BadNets, we also test its all-to-all (A2A) attack setting, in which the target labels $y_t$ are set to all labels by using $y_t = (y + 1)\%C$ (where $\%$ represents the modulo operation). The triggers for BadNets and CLA are set to randomly generated patterns with size $3 \times 3$. For Blended, we use random noise patterns as the triggers. Note that CLA doesn't poison labels, and their poisoning rate is set to $8\%$, which is $80\%$ of the images of the target class. For other attacks, the poisoning rate is set to $10\%$.

To assess the robustness of our method, we conducted additional evaluations against two advanced out-of-setting attacks, namely WaNet (Nguyen & Tran, 2021) and IAB (Nguyen & Tran, 2020). Here, "out-of-setting" refers to these attacks involving control over the training process, which falls outside the feasible setting of our method.

**Defenses.** Defense aims to obtain a clean model without backdoor behaviors. We assume that the defender has access to the poisoned training set and compare our approaches with entropy-based pruning (EP) (Zheng et al., 2022b) and channel Lipschitzness-based pruning (CLP) (Zheng et al., 2022a). In addition, we also compare our approach with other methods that require an extra clean dataset, i.e., adversarial neuron pruning (ANP) (Wu & Wang, 2021) and fine-pruning (FP) (Liu et al., 2018).

**Metrics.** We evaluate the effectiveness of different methods using the attack clean accuracy (ACC) and attack success rate (ASR). The ACC for a given model $F$ is given by the indicator function $\mathbb{I}$ in the sum of all samples $(\boldsymbol{x}, y) \in \mathcal{D}_{\text{test}}$ such that $\arg\max(F(\boldsymbol{x})) = y$. The ASR is calculated in the same way, but with the attack target label $y_t$ instead of the true label $y$ and with $\delta(\boldsymbol{x})$ instead of $\boldsymbol{x}$. The ACC measures the performance of the model on benign samples, while ASR reflects the degree of backdoor behavior retained in the model. Therefore, our goal is to reduce ASR while keeping ACC from dropping too much.

Table 2: Comparison of the proposed DRR with three state-of-the-art backdoor defenses on CIFAR-10. The best results are denoted with bold font and the second best with an underline.

| Model | Attack | Backdoored | | FP | | ANP | | CLP | | EP | | DRR | |
|---|---|---|---|---|---|---|---|---|---|---|---|---|---|
| | | ACC | ASR | ACC | ASR | ACC | ASR | ACC | ASR | ACC | ASR | ACC | ASR |
| ResNet18 | WaNet | 94.09 | 99.82 | 89.29 | 2.09 | 92.96 | 62.64 | 92.52 | 11.28 | 92.37 | **0.56** | 93.53 | 1.44 |
| | IAB | 93.97 | 99.30 | 90.82 | 20.87 | 91.47 | 2.81 | 92.66 | 7.60 | 91.73 | 1.64 | **93.26** | 1.51 |
| | Average | 93.03 | 99.56 | 90.06 | 11.48 | 92.22 | 32.73 | 92.59 | 9.44 | 92.05 | **1.10** | 93.39 | 1.48 |
| WRN-28-1 | WaNet | 89.46 | 95.45 | 86.40 | 5.40 | 85.63 | **1.98** | 86.67 | 4.24 | 80.94 | 9.06 | **87.47** | 2.57 |
| | IAB | 88.91 | 82.75 | 85.03 | 23.23 | 84.78 | 2.94 | 87.42 | 0.84 | **87.78** | 2.60 | 87.02 | **0.46** |
| | Average | 89.19 | 89.10 | 85.72 | 14.32 | 85.21 | 2.46 | 87.05 | 2.54 | 84.36 | 5.83 | **87.25** | 1.52 |
| Average | | 91.61 | 94.33 | 87.89 | 12.90 | 88.71 | 17.59 | 89.82 | 5.99 | 88.21 | 3.47 | **90.32** | **1.50** |

Table 3: Comparison of the proposed DRR with three state-of-the-art backdoor defenses on CIFAR-10 with lower poisoning ratios: 5.0% (left, 4.0% for CLA) and 1.0% (right).

| Model | Attack | Backdoored | | DRR | |
|---|---|---|---|---|---|
| | | ACC | ASR | ACC | ASR |
| ResNet-18 | BadNets | 94.14 | 99.99 | 93.90 | 1.58 |
| | BadNets(A2A) | 94.18 | 84.71 | 93.76 | 0.94 |
| | Blended | 94.81 | 100.00 | 94.01 | 1.73 |
| | CLA | 94.86 | 89.94 | 94.78 | 0.94 |
| | Average | 94.50 | 93.66 | 94.11 | 1.30 |
| WideResNet-28-1 | BadNets | 92.22 | 99.99 | 90.14 | 1.51 |
| | BadNets(A2A) | 92.17 | 91.26 | 92.06 | 1.70 |
| | Blended | 91.72 | 93.76 | 89.89 | 0.70 |
| | CLA | 92.87 | 36.23 | 91.44 | 4.63 |
| | Average | 92.25 | 80.31 | 90.88 | 2.14 |

| Model | Attack | Backdoored | | DRR | |
|---|---|---|---|---|---|
| | | ACC | ASR | ACC | ASR |
| ResNet-18 | BadNets | 94.83 | 98.79 | 94.69 | 0.97 |
| | BadNets(A2A) | 94.81 | 86.52 | 94.52 | 0.66 |
| | Blended | 95.03 | 99.99 | 94.88 | 1.84 |
| | CLA | 94.82 | 16.18 | 94.86 | 6.34 |
| | Average | 94.87 | 75.37 | 94.74 | 2.45 |
| WideResNet-28-1 | BadNets | 92.46 | 99.92 | 90.74 | 4.41 |
| | BadNets(A2A) | 92.43 | 79.94 | 91.46 | 1.47 |
| | Blended | 92.55 | 99.80 | 89.89 | 0.70 |
| | CLA | 92.39 | 3.49 | 92.39 | 3.49 |
| | Average | 92.46 | 70.79 | 91.12 | 2.52 |

## 5.2 QUANTITATIVE RESULTS

Table 1 highlights the exceptional performance of DRR relative to earlier methods in diverse scenarios. On average, employing DRR results in a substantial three-fold decrease in ASR compared to EP, the leading pruning-based defense method to date. Although FP shows satisfactory performance, it experiences significant reductions in clean accuracy. Conversely, ANP incurs lesser degradation in clean accuracy compared to FP, but it proves ineffective in robustly countering backdoor attacks, especially under BadNets(A2A) and CLA conditions. CLP markedly lowers ASR to nearly zero in most situations, while keeping a decent level of accuracy on unadulterated data. Nevertheless, its efficacy against the Blended attack is lacking, potentially due to its assumption of triggers being minor disturbances, whereas in Blended attacks, triggers span the entire image. While EP generally holds up well, it does not offer ample defense against Blended attacks on CIFAR-10 with WRN-28-1. In contrast, our proposed DRR significantly diminishes ASR in the majority of cases, underscoring its robustness and efficacy.

In Table 3, we present the outcomes of our experimental evaluations at reduced poisoning ratios. The table on the left delineates the performance metrics obtained with a poisoning ratio of 5.0% (modified to 4.0% for CLA specifically), whereas the table on the right corresponds to a significantly lower poisoning ratio of 1.0%. Despite the reduced presence of poisoning data—a critical element in determining the efficacy of our method—DRR demonstrates a consistent and robust performance, even in scenarios characterized by minimal poisoning ratios.

## 5.3 EMPIRICAL JUSTIFICATION FOR ASSUMPTION 2

In this subsection, the practical validity of the central assumption in our methodology is substantiated, specifically referring to assumption 2. Using ResNet-18 as a representative example, the $C$ value is tested across all layers under four distinct attacks. The empirical $C$ is identified as the ratio of the norm of expectation difference to the maximum eigenvalue of the covariance matrices of both benign and poisoned features. As illustrated in Figure 2, nearly all layers exhibit a consistent $C > 5$, a sufficiently large threshold ensuring a close approximation to the expectation difference. This observation underscores the solid empirical basis of our assumption. By its definition, $C$ symbolizes the extent of separation between benign and poisoned features. Observations also highlight varying $C$ distributions across different attacks. For instance, BadNets predominantly show larger $C$ values in superficial layers, CLA in intermediate layers, and Blended in certain deeper layers exclusively.

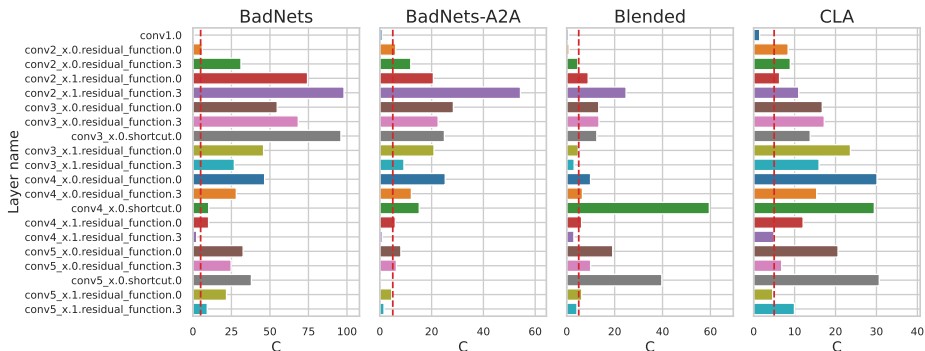

Figure 2: The visualization of the empirical constant $C$ across all layers of ResNet-18 under four different attacks.

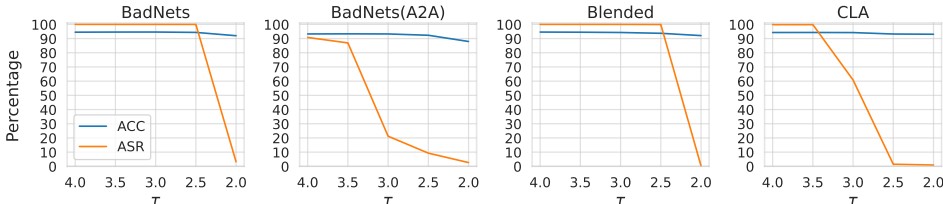

Figure 3: ACC and ASR of DRR with varying threshold.

## 5.4 IMPACT OF THRESHOLD SELECTION

To evaluate the influence of threshold selection on the effectiveness of our proposed approach, experiments were carried out on the CIFAR-10 dataset using the ResNet-18 architecture. Four attack methods were explored, as shown in Figure 3. The results reveal that the ASR consistently begins to substantially decline to almost zero when the threshold, $\tau$, is set to 2 across all attack types.

## 6 DISCUSSION

**Adaptation of existing pruning-based methods.** This research broadens the field of neuron pruning to include a more extensive rank reduction operation, leading to an engaging query: is it possible to modify existing pruning-based methods to their corresponding "rank reduction" versions? Theoretically speaking, it is plausible as each method generally employs a criterion to choose the specific neurons for pruning. For example, fine-pruning selects neurons with the lowest mean activation values. This can be adapted into our framework by opting for the direction with the smallest projected mean. Likewise, in ANP, the initial step is the optimization of an adversarial perturbation along the neuron direction. To integrate it into our framework, optimization of the adversarial perturbations along all possible directions is essential. Investigating additional criteria for use within our framework presents a fascinating path for upcoming research.

**Limitations.** Despite its wider applicability, DRR introduces additional memory cost as full batch gradient descent is needed to ensure a quick and accurate solution when optimizing the target direction. This demand increases particularly with larger datasets, contributing to substantial memory usage. In our implementation, to strike a balance between time and memory usage, we retrieve the features of the entire dataset each time a new layer is processed. This makes the processing time scaled linearly w.r.t. the number of layers. We leave it for future work to accelerate the algorithm.

## 7 RELATED WORK

### 7.1 BACKDOOR ATTACKS

**Backdoor attacks** emerge in the seminal work of BadNets (Gu et al., 2019), wherein a small set of targeted label-flipped data with a specific trigger is injected into a training set, resulting in misclassification when predicting samples with the trigger. Subsequent works, such as Blended Attack (Chen et al., 2017), blended the trigger pattern to make it more invisible to human beings. Reflection Backdoor (Refool) (Liu et al., 2020) utilized the form of natural reflection for trigger design. Clean Label Attack (CLA) (Turner et al., 2019) perturbed the input image while keeping its content consistent with the target label, enabling the model to better memorize the trigger pattern while remaining imperceptible to human beings. In order to make the detection of backdoors even more challenging, dynamic backdoor attacks have been developed, such as Input-Aware Backdoor (IAB) (Nguyen & Tran, 2020), Warping-based Backdoor (WaNet) (Nguyen & Tran, 2021), and Sample Specific Backdoor Attack (SSBA) (Li et al., 2021). These dynamic attacks generate a unique trigger for each input, rendering traditional defense mechanisms less effective.

### 7.2 BACKDOOR DEFENSE

Backdoor defenses mainly consists of two paradigms: *post-training* and *in-training* processing.

**Post-training processing** aims to mitigate the backdoor threat after a model has been trained. Neural Cleanse (NC) (Wang et al., 2019) reverse-engineers backdoor triggers and retrains the model to remove the backdoor. Neural Attention Distillation (NAD) (Li et al., 2020a) incorporates knowledge distillation methods, while Mode Connectivity Repair (MCR) (Zhao et al., 2020) eliminates the trigger effect. Another notable research direction against backdoor attacks is neuron pruning, which involves selectively removing neurons associated with backdoor activity. This can be traced back to the notion of fine-pruning as proposed by (Liu et al., 2018), which argued that neurons related to backdoor activity should demonstrate a low magnitude of activation when presented with benign features. ANP (Wu & Wang, 2021) argued that neurons relating to the prediction of backdoor samples should display a heightened level of sensitivity to adversarial perturbations. Channel Lipschitzness-based Pruning (Zheng et al., 2022a) measured the sensitivity of neurons to input image perturbation by calculating the Lipschitz constant of each neuron. Entropy-based pruning (Zheng et al., 2022b) observed that backdoor neurons should have significant bi-Gaussian distributions formed by benign and poisoned populations, and such distributions should display reduced entropy compared to normal neurons, which generally present as a unimodal distribution.

**In-training processing** prevents backdoors from being injected during training. These methods leverage the distinct distributions of poisoned and clean data in the feature space (Huang et al., 2021; Li et al., 2023) to identify and mitigate the presence of backdoors. Robust statistics approaches (Hayase & Kong, 2020; Tran et al., 2018), which include input perturbation techniques (Gao et al., 2019; Doan et al., 2020) and semi-supervised training (Huang et al., 2021), aim to filter out the poisoned data from the training set. Stronger data augmentation techniques (Borgnia et al., 2021), such as CutMix (DeVries & Taylor, 2017), CutOut (DeVries & Taylor, 2017), and MaxUp (Gong et al., 2020), have also been proposed to suppress the effects of backdoor attacks. One of the most recent approaches achives clean training on poisoned dataset via adaptively filtering (Gao et al., 2023).

## 8 CONCLUSION

In this work, we explore the inadequacies of existing pruning-based defense: it requires excessive pruning when the trigger effect entangles various neurons. To address this limitation, we present a Directional Rank Reduction (DRR) framework, which is an extension of neuron pruning. DRR suppresses neuron pruning by making it a more flexible rank reduction operation along arbitrary direction. DRR is well justified by theory, and we validate its superiority as compared to existing pruning-based methods via exhaustive experiments. Our work advances the state-of-the-art in backdoor defense, which could shed light on future research.

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

## A  OMITTED PROOF

**Theorem 2.** *Given a two-component Gaussian mixture model with the following probability density function:*

$$p(\boldsymbol{X}) = \pi_1 \mathcal{N}(\boldsymbol{X}|\boldsymbol{\mu}_1, \Sigma_1) + \pi_2 \mathcal{N}(\boldsymbol{X}|\boldsymbol{\mu}_2, \Sigma_2),$$

*where $\mathcal{N}(\boldsymbol{X}|\mu, \Sigma)$ represents the multivariate normal distribution with mean $\mu$ and covariance matrix $\Sigma$. Let $\boldsymbol{d} = \boldsymbol{\mu}_1 - \boldsymbol{\mu}_2$ denotes the direction of the mean difference and $\boldsymbol{v}$ denotes an arbitrary direction. Under assumption 2, a lower bound of the third central moment is given by:*

$$\mathbb{E}[(\boldsymbol{v}^T(\boldsymbol{X} - \boldsymbol{\mu}))^3] \geq \pi_1 \pi_2 (\boldsymbol{d}^T \boldsymbol{d}) \left(\boldsymbol{v}^T \left(3\Sigma_1 + 3\Sigma_2 + 2\boldsymbol{d}\boldsymbol{d}^T\right) \boldsymbol{v}\right),$$

*and maximizing the lower bound gives:*

$$\cos(\boldsymbol{v}, \boldsymbol{d}) \geq \sqrt{C^2 - 9}/C.$$

*Proof.* Let $\mathbf{d} = \boldsymbol{\mu}_1 - \boldsymbol{\mu}_2$ represent the direction between the two Gaussian means. Also, let $\boldsymbol{\mu}$ represent the overall mean of the mixture model, given by:

$$\boldsymbol{\mu} = \pi_1 \boldsymbol{\mu}_1 + \pi_2 \boldsymbol{\mu}_2$$

The third central moment along the direction $\boldsymbol{v}$ for the mixture model can be represented as:

$$\mathbb{E}[(\boldsymbol{v}^T(\boldsymbol{X} - \boldsymbol{\mu}))^3] = \pi_1 \mathbb{E}_{\boldsymbol{\mu}_1}[(\boldsymbol{v}^T(\boldsymbol{X} - \boldsymbol{\mu}))^3] + \pi_2 \mathbb{E}_{\boldsymbol{\mu}_2}[(\boldsymbol{v}^T(\boldsymbol{X} - \boldsymbol{\mu}))^3]$$

Expanding for the Gaussian centered at $\boldsymbol{\mu}_1$, we have:

$$\mathbb{E}_{\boldsymbol{\mu}_1}[(\boldsymbol{v}^T(\boldsymbol{X} - \boldsymbol{\mu}))^3] = \int (\boldsymbol{v}^T(x - \boldsymbol{\mu}))^3 N(x|\boldsymbol{\mu}_1, \Sigma_1) dx$$

Using the change of variable $x = \boldsymbol{\mu}_1 + z$, we rewrite the integral as:

$$\int (\boldsymbol{v}^T(z + \boldsymbol{\mu}_1 - \boldsymbol{\mu}))^3 N(z|0, \Sigma_1) dz$$

Expanding the cube and noting that odd powers of $z$ (centered around zero) will integrate to zero due to the symmetry of the Gaussian, we have:

$$\int (\boldsymbol{v}^T(\boldsymbol{\mu}_1 - \boldsymbol{\mu}))^3 N(z|0, \Sigma_1) dz$$
$$= \int \boldsymbol{v}^T z^3 + 3\boldsymbol{v}^T z^2 (\boldsymbol{\mu}_1 - \boldsymbol{\mu}) + 3\boldsymbol{v}^T z (\boldsymbol{\mu}_1 - \boldsymbol{\mu})^2 + (\boldsymbol{v}^T(\boldsymbol{\mu}_1 - \boldsymbol{\mu}))^3 dz$$
$$= 3(\boldsymbol{v}^T \Sigma_1 \boldsymbol{v}) \boldsymbol{v}^T (\boldsymbol{\mu}_1 - \boldsymbol{\mu}) + (\boldsymbol{v}^T(\boldsymbol{\mu}_1 - \boldsymbol{\mu}))^3$$

where:

$$\begin{aligned}
\boldsymbol{v}^T(\boldsymbol{\mu}_1 - \boldsymbol{\mu}) &= \boldsymbol{v}^T(\boldsymbol{\mu}_1 - \pi_1 \boldsymbol{\mu}_1 + \pi_2 \boldsymbol{\mu}_2) \\
&= \boldsymbol{v}^T((1 - \pi_1)\boldsymbol{\mu}_1 + \pi_2(\boldsymbol{d} - \boldsymbol{\mu}_1)) \\
&= \pi_2 \boldsymbol{v}^T \boldsymbol{d}
\end{aligned}$$

Then:

$$3(\boldsymbol{v}^T \Sigma_1 \boldsymbol{v}) \boldsymbol{v}^T(\boldsymbol{\mu}_1 - \boldsymbol{\mu}) + (\boldsymbol{v}^T(\boldsymbol{\mu}_1 - \boldsymbol{\mu}))^3$$
$$= 3\pi_2 (\boldsymbol{v}^T \Sigma_1 \boldsymbol{v})(\boldsymbol{v}^T \boldsymbol{d}) + (\boldsymbol{v}^T \boldsymbol{d})^3$$

The same expansion can be applied to the second Gaussian, then we have:

$$
\begin{aligned}
\mathbb{E}[(\boldsymbol{v}^T(\boldsymbol{X}-\boldsymbol{\mu}))^3] &= \pi_1\mathbb{E}_{\boldsymbol{\mu}_1}[(\boldsymbol{v}^T(\boldsymbol{X}-\boldsymbol{\mu}))^3] + \pi_2\mathbb{E}_{\boldsymbol{\mu}_2}[(\boldsymbol{v}^T(\boldsymbol{X}-\boldsymbol{\mu}))^3] \\
&= 3\pi_1\pi_2(\boldsymbol{v}^T\Sigma_1\boldsymbol{v})(\boldsymbol{v}^T\boldsymbol{d}) + (\boldsymbol{v}^T\boldsymbol{d})^3 + 3\pi_1\pi_2(\boldsymbol{v}^T\Sigma_2\boldsymbol{v})(\boldsymbol{v}^T\boldsymbol{d}) + (\boldsymbol{v}^T\boldsymbol{d})^3 \\
&= \pi_1\pi_2(\boldsymbol{v}^T\boldsymbol{d})\left(\boldsymbol{v}^T\left(3\Sigma_1 + 3\Sigma_2 + 2\boldsymbol{d}\boldsymbol{d}^T\right)\boldsymbol{v}\right) \\
&= \pi_1\pi_2\left(\left(\boldsymbol{d}^T\boldsymbol{d}\right)\left(\boldsymbol{v}^T\left(3\Sigma_1 + 3\Sigma_2 + 2\boldsymbol{d}\boldsymbol{d}^T\right)\boldsymbol{v}\right) + \left((\boldsymbol{v}-\boldsymbol{d})^T\boldsymbol{d}\right)\left(\boldsymbol{v}^T\left(3\Sigma_1 + 3\Sigma_2 + 2\boldsymbol{d}\boldsymbol{d}^T\right)\boldsymbol{v}\right)\right) \\
&\geq \pi_1\pi_2\left(\left(\boldsymbol{v}^T\left(3\Sigma_1 + 3\Sigma_2 + 2\boldsymbol{d}\boldsymbol{d}^T\right)\boldsymbol{v}\right) - \left(\boldsymbol{v}^T\left(3\Sigma_1 + 3\Sigma_2 + 2\boldsymbol{d}\boldsymbol{d}^T\right)\boldsymbol{v}\right)\right) \\
&\geq \pi_1\pi_2\left(\boldsymbol{v}^T\left(3\Sigma_1 + 3\Sigma_2 + 2\boldsymbol{d}\boldsymbol{d}^T\right)\boldsymbol{v}\right)
\end{aligned}
$$

This gives us an lower bound on the third central moment for an arbitrary direction $\boldsymbol{v}$. We then formulate the optimization problem to maximize the above expression:

$$
\arg\max_{\boldsymbol{v}} \ \boldsymbol{v}^T\left(3\Sigma_1 + 3\Sigma_2 + 2\boldsymbol{d}\boldsymbol{d}^T\right)\boldsymbol{v}
$$

The solution to this problem is the top eigenvector of the inner component of the quadratic form $3\Sigma_1 + 3\Sigma_2 + 2\boldsymbol{d}\boldsymbol{d}^T$, which can be seen as a perturbation to the matrix $\boldsymbol{d}\boldsymbol{d}^T$. According to Davis-Kahan $\sin\Theta$ theorem, the angle between the top eigenvector before and after perturbation can be represented by:

$$
\sin(\boldsymbol{v}, \boldsymbol{d}) \leq \frac{\|3\Sigma_1 + 3\Sigma_2\|_2}{\mathrm{gap}(2\boldsymbol{d}\boldsymbol{d}^T + 3\Sigma_1 + 3\Sigma_2)} \leq \frac{3\|\Sigma_1 + \Sigma_2\|_2}{2\|\boldsymbol{d}\|_2^2},
$$

where $\|\cdot\|_2$ is the spectral norm and $\|\cdot\|_2^2$ is the squared $L2$ norm. According to the separation dominance assumption, let $\lambda = max(\lambda_{max}(\Sigma_1), \lambda_{max}(\Sigma_2))$ we have:

$$
\sin(\boldsymbol{v}, \boldsymbol{d}) \leq 3/C,
$$

Hence:

$$
\cos(\boldsymbol{v}, \boldsymbol{d}) \geq \sqrt{1 - \sin(\boldsymbol{v}, \boldsymbol{d})} = \sqrt{C^2 - 9}/C
$$

$\square$

