# OpenReview forum: "Directional Rank Reduction for Backdoor Defense"
_ICLR.cc/2024/Conference — Submitted to ICLR 2024_

### Official Review · Reviewer_X3hw · 2023-10-30

**Soundness:** 2 fair
**Presentation:** 3 good
**Contribution:** 2 fair
**Rating:** 5
**Confidence:** 3

**Summary:**

This paper argues that existing pruning-based defense methods can be ineffective at times and introduces Directional Rank Reduction (DRR) to identify toxic directions. In this study, the method approximates the target direction by maximizing the third central moment, supported by rigorous theoretical justification, and constructs a projection matrix to eliminate the toxic direction. DRR demonstrated outstanding performance in terms of both accuracy (ACC) and adversarial success rate (ASR).

**Strengths:**

1. This study shows an interesting finding that the backdoor trigger effects are not always aligned with fixed dimensions of the feature space, pruning-based methods are usually ineffective.
2. The proposed DRR method performed well on both ACC and ASR compared to other methods.

**Weaknesses:**

1. In the first equation on Page 3, it seems feasible to do the defense by reducing the norm of the residual matrix to align the benign and poisoned features seems feasible. The features from benign examples move towards the backdoored features. Does the movement hurt the model's clean performance?

2. The last equation on Page 4 has a strong assumption that all the clean examples are centered around the mean of them. Namely, the method assumes that the distances from all the clean examples to the example center are the same. The examples marked as yellow in Figure 1 are distributed like a circle. However, the real-world data distribution often deviates from the assumption. The distribution could be elliptical-like. In this case, the obtained v is not optimal anymore.

3. In the third row of Table 2, DRR achieves a better trade-off. Why it demonstrates a higher accuracy (ACC) instead of a lower ASR?

4. This approach requires the optimization of a vector in each layer, which could be expensive.

minor: All the equations are not numbered!

**Questions:**

1. "How the direction vector v is initialized in the paper, and do different initialization methods lead to varying results?

2. In Figure 2, the value of C for certain layers is not significant. Is it possible to skip some layers when computing v?

---

> ### Author Response · Authors · 2023-11-21
>
> ### Weakenesses:
>
> #### W1: In the first equation on Page 3, it seems feasible to do the defense by reducing the norm of the residual matrix to align the benign and poisoned features seems feasible. The features from benign examples move towards the backdoored features. Does the movement hurt the model's clean performance?
>
> Answer: As we points out in our paper, rank reduction is an extension of neuron pruning (which is also a modification of the weight matrix). Established neuron pruning techniques, such as CLP and EP, have been demonstrated to maintain model performance effectively post-pruning. Our rank reduction, which 1) remove only one rank (instead of multiple ranks in neuron pruning) and 2) remove more targetedly to the direction that related to the backdoor behavior should intuitively affect less to the performance than pruning methods in general. Empirically, we show that our method affects the less on the performance compare to the previous method.
>
> #### W2: The last equation on Page 4 has a strong assumption that all the clean examples are centered around the mean of them. Namely, the method assumes that the distances from all the clean examples to the example center are the same. The examples marked as yellow in Figure 1 are distributed like a circle. However, the real-world data distribution often deviates from the assumption. The distribution could be elliptical-like. In this case, the obtained v is not optimal anymore.
>
> Answer: We appreciate the opportunity to clarify a potential misunderstanding highlighted by the reviewer regarding the assumptions underlying our method. Contrary to the reviewer's interpretation, our approach does not assume uniform distances of all clean examples from their mean, nor does it presuppose a circular distribution of data points as depicted in Figure 1. The shape of the ellipse in a multivariate Gaussian distribution is determined by its covariance matrix. Only when the covariance matrix is *isotropic*, which means the variance along all directions are the same, the data is distributed like a circle as mentioned by the reviewer. However, we *do not* put any constraints on whether the covariance is isotropic or not. The only assumption about the covariance matrix we made is assumption 2, which limits the maximum variance of the data distribution. Hence, an elliptical-like distribution is considered in our method, where the Gaussian distribution has a covariance matrix that is not isotropic.
>
> The equation on page 4, which the reviewer refers to, formulates an optimization problem. The objective of this problem is to determine an optimal unit direction vector that maximizes the third central moment when data is projected onto this vector. The stipulation that the vector be of unit length is a constraint applied to the direction vector itself, rather than to the data samples. This is a standard practice in such optimization problems to ensure the direction vector is normalized and hence, the focus is on the direction rather than the magnitude.
>
> #### W3: In the third row of Table 2, DRR achieves a better trade-off. Why it demonstrates a higher accuracy (ACC) instead of a lower ASR?
>
>
> Answer: We appreciate the reviewer's insightful observation and recognize the necessity of providing a clearer explanation in our manuscript. The comparatively minimal impact on model performance observed in our study can be attributed to our method's strategy of removing at most one rank per layer. This approach is considerably more conservative than traditional neuron pruning methods, which often entail the removal of multiple ranks. If the number of ranks is tuned to more according to specific scenario, DRR can achieve even lower ASR.
>
>
>
> #### W4: This approach requires the optimization of a vector in each layer, which could be expensive.
>
> Answer: We appreciate the inquiry regarding the computational efficiency of our approach.
> Primarily, the optimization process can be parallelized across different network layers, offering a substantial decrease in the required time for optimization. Furthermore, in scenarios with high feature dimensions or large dataset sizes, dimensionality reduction through Principal Component Analysis (PCA) can be employed prior to the directional learning phase. This step effectively reduces the computational burden. Subsequently, the learned direction within this reduced space is projected back onto the original space, thereby economizing on memory and computational demands while preserving the integrity of the optimization process.

---

> > ### Author Response · Authors · 2023-11-21
> >
> > ### Questions:
> >
> > #### Q1: How the direction vector v is initialized in the paper, and do different initialization methods lead to varying results?
> >
> > Answer: The reviewer's comment is well-taken. Within our manuscript, we initialize the vector v as the standard basis vector that corresponds to the maximal third central moment upon projection of the data. To thoroughly examine the impact of various initialization strategies, we have implemented two distinct approaches: (1) random initialization, and (2) initialization using the original standard basis vector that exhibits the highest third central moment. The findings from these methodologies are presented in the following section:
> >
> > |                 |              | Backdoored |       | DRR (random) |        | DRR (original) |      |
> > |-----------------|--------------|------------|-------|--------------|--------|----------------|------|
> > |                 |              | ACC        | ASR   | ACC          | ASR    | ACC            | ASR  |
> > | ResNet-18       | BadNets      | 94.83      | 98.79 | 92.20        | 1.87   | 92.01          | 3.18 |
> > |                 | BadNets(A2A) | 94.81      | 86.52 | 93.11        | 1.30   | 92.42          | 1.93 |
> > |                 | Blended      | 95.03      | 99.99 | 90.57        | 3.42   | 94.40          | 2.31 |
> > |                 | CLA          | 94.82      | 16.18 | 93.60        | 0.96   | 93.40          | 0.98 |
> > |                 | Average      | 94.87      | 75.37 | 92.37        | 1.89   | 93.06          | 2.10 |
> > | WideResNet-28-1 | BadNets      | 92.46      | 99.92 | 89.97        | 2.34   | 89.56          | 1.69 |
> > |                 | BadNets(A2A) | 92.43      | 79.94 | 88.60        | 1.80   | 91.94          | 1.29 |
> > |                 | Blended      | 92.55      | 99.80 | 91.32        | 100.00 | 91.50          | 2.74 |
> > |                 | CLA          | 92.39      | 3.49  | 90.57        | 1.07   | 91.41          | 6.19 |
> > |                 | Average      | 92.46      | 70.79 | 90.12        | 26.30  | 91.10          | 2.98 |
> >
> > The results indicate that different intialization work well in different scenarios. However, in general, initializing the vector with original standard basis with the highest criterion performs more robust than random initialization.
> >
> > #### Q2: In Figure 2, the value of C for certain layers is not significant. Is it possible to skip some layers when computing v?
> >
> >
> > Answer: The reviewer's suggestion is indeed a compelling proposition. In this way we can reduce the computational cost and the hurt to the normal performance of the model. However, currently the exact calculation of C requires the access to both poisoned data and benign data separately, which is impractice in our scenario. Nonetheless, we acknowledge the potential benefits of such a strategy and consider it an intriguing avenue for future exploration.

---

> ### Comment · Reviewer_X3hw · 2023-12-01
> **Response to rebuttal**
>
> Thank the authors for providing a detailed rebuttal. However, I still have concerns regarding the motivation of the method.
>
> Figure 1 illustrates why the proposed method works, which is only valid when the data distribution is circle-like. In real-world datasets, the distribution is not the case. Namely, the motivation or the explanation of the proposed method is not convincing anymore.
>
> I understand that the authors do not make any assumptions about the data explicitly. But please check the motivation illustrated in Figure 1. The illustration does not make sense for real-world data distribution.
>
> Hence, I tend to keep my original score.

---

### Official Review · Reviewer_1xjT · 2023-10-31

**Soundness:** 3 good
**Presentation:** 3 good
**Contribution:** 3 good
**Rating:** 6
**Confidence:** 4

**Summary:**

This paper proposes a novel backdoor defense method, which utilizes rank reduction to mitigate backdoor in the model. The idea of rank reduction is interesting and brings a new insight into the area.

**Strengths:**

1. The idea is novel and provides a new insight.
2. This paper is technically sound and easy to follow.
3. The experimental results demonstrate its effectiveness in backdoor defense.

**Weaknesses:**

1.Although this work is interesting, it has a limitation. This paper assumes the defender can get access to the backdoored image. However, this is hard to get in actual situations and thus limits its use greatly. I wonder whether it works without these backdoored data.
2. The backdoor attacks that this paper test is not enough. I suggest the authors to test the newest input-specific backdoor attacks in 2022. It's important to identify whether this method can achieve SOTA.

**Questions:**

1.Does it work without the attacker's backdoored data?

---

> ### Author Response · Authors · 2023-11-21
>
> We would like to thank the reviewer for their appreciation of our work. Below, we have provided our response to the reviewer's concerns.
>
> ### Weakenesses:
>
> #### W1: Although this work is interesting, it has a limitation. This paper assumes the defender can get access to the backdoored image. However, this is hard to get in actual situations and thus limits its use greatly. I wonder whether it works without these backdoored data.
>
> Answer: First, to answer the reviewer's final question, the rank reduction framework can be adpated to *any* scenarios with or *without* backdoored data, but the metric we adopt to obtain the direction, i.e., third central moment, requires the access to the backdoored data. If other metrics is later being invented to obtain the direction, then the method could be backdoored data-free.
>
> Second, we want to argue that the scenario in which the defender has access to the full dataset is a prevalent assumption within the backdoor attack research community, exemplified by the concept of adopting a third-party dataset as discussed in [1]. Some of methodologies that employ this setting are outlined in: [2, 3, 4, 5].
>
> [1] Li, Y., Jiang, Y., Li, Z. and Xia, S.T., 2022. Backdoor learning: A survey. IEEE Transactions on Neural Networks and Learning Systems.
> [2] Chen, B., Carvalho, W., Baracaldo, N., Ludwig, H., Edwards, B., Lee, T., Molloy, I. and Srivastava, B., Detecting Backdoor Attacks on Deep Neural Networks by Activation Clustering.
> [3] Zheng, R., Tang, R., Li, J. and Liu, L., 2022. Pre-activation Distributions Expose Backdoor Neurons. Advances in Neural Information Processing Systems, 35, pp.18667-18680.
> [4] Tran, B., Li, J. and Madry, A., 2018. Spectral signatures in backdoor attacks. Advances in neural information processing systems, 31.
> [5] Hayase, J., Kong, W., Somani, R. and Oh, S., 2021, July. Spectre: Defending against backdoor attacks using robust statistics. In International Conference on Machine Learning (pp. 4129-4139). PMLR.
>
> #### W2: The backdoor attacks that this paper test is not enough. I suggest the authors to test the newest input-specific backdoor attacks in 2022. It's important to identify whether this method can achieve SOTA.
>
> Answer: Actually, the experiments with IAB and WaNet, which are both input-specific backdoor attacks, are in the paper. Please refer to Table 2 in our submitted paper to see our results tested on input-aware dynamic attack (IAB) and Warping-based backdoor attack (WaNet).
>
> Moreover, we conduct more experiments on other types of attacks (AdaptiveBlend, SIG and Smooth), where the results is as shown below:
> |                 |               | Backdoored |        | FP    |       | ANP   |       | EP    |       | CLP   |       | DRR   |      |
> |-----------------|---------------|------------|--------|-------|-------|-------|-------|-------|-------|-------|-------|-------|------|
> |                 |               | ACC        | ASR    | ACC   | ASR   | ACC   | ASR   | ACC   | ASR   | ACC   | ASR   | ACC   | ASR  |
> | ResNet-18       | AdaptiveBlend | 94.79      | 100.00 | 89.47 | 5.29  | 82.18 | 0.30  | 94.43 | 1.74  | 93.68 | 33.52 | 90.25 | 3.79 |
> |                 | SIG           | 94.01      | 98.22  | 88.94 | 45.70 | 89.38 | 2.36  | 87.36 | 30.84 | 89.75 | 94.28 | 87.10 | 0.07 |
> |                 | Smotth        | 94.59      | 100    | 87.12 | 100   | 92.65 | 81.23 | 94.24 | 3.99  | 87.24 | 89.03 | 94.03 | 3.58 |
> | WideResNet-28-1 | AdaptiveBlend | 92.37      | 100.00 | 84.77 | 51.88 | 82.06 | 42.70 | 90.45 | 5.18  | 84.40 | 74.57 | 91.13 | 0.86 |
> |                 | SIG           | 84.03      | 96.20  | 82.65 | 5.22  | 81.37 | 0.00  | 83.82 | 0.00  | 84.04 | 0.00  | 82.85 | 0.00 |
> |                 | Smotth        | 92.19      | 100    | 84.52 | 6.32  | 89.98 | 100   | 91.45 | 8.78  | 91.29 | 9.03  | 91.88 | 2.74 |

---

### Official Review · Reviewer_niUP · 2023-10-31

**Soundness:** 3 good
**Presentation:** 2 fair
**Contribution:** 3 good
**Rating:** 5
**Confidence:** 4

**Summary:**

The paper presents a fascinating new method for backdoor defense in neural networks. The key idea of projecting the "toxic direction" that maximizes the difference between clean and poisoned features is novel and seems promising.

The theoretical analysis provides valuable insights into the limitations of standard neuron pruning approaches. Framing the problem as rank reduction along arbitrary directions rather than fixed neuron directions is a significant conceptual shift.

**Strengths:**

1. The idea of maximizing the third central moment is enjoyable. This idea yields a novel insight.
2. The connection between neuron pruning and rank reduction is also an exciting topic.
3. The visualization of the separation constant C provides good justification for the theoretical assumptions.

**Weaknesses:**

1.	More experiments can be conducted (BadNet, Blended, CLA, WaNet, and IAB are insufficient.) The authors can consider attacks like SIG [1] and low frequency (Smooth) [2]. Since your method also took latent separability as an assumption, Adapt-blend and Adapt-patch attacks [3] should also be considered. Evaluating robustness to adaptive attacks that try to evade the defense would be useful to understand limitations.
2.	The references and notations should be clarified. For example, what is the reference to Proposition 1?
3.	Also, the readability and organization of this paper need to be improved. It is better if an algorithm is provided.

[1] A new backdoor attack in cnns by training set corruption ICIP 2019

[2] Rethinking the Backdoor Attacks’ Triggers: A Frequency Perspective ICCV2021

[3] Revisiting the Assumption of Latent Separability for Backdoor Defenses, ICLR 2023

**Questions:**

1.	The memory and computational complexity could be analyzed more thoroughly, especially how the approach scales with larger datasets/models. Are there ways to make the optimization more efficient?
3.	How many extension directions v_i have you used?
4.	Modifying the weight matrix may cause a performance drop in many cases. How can your projection keep the performance?
5.	The proof needs to be more rigorous. Why use the consequence of the proof in the middle of the proof?

---

> ### Author Response · Authors · 2023-11-21
>
> We would like to thank the reviewer for the detailed review. We will make changes based on the feedback. Please see our responses:
>
> ### Weaknesses:
>
> #### W1: More experiments can be conducted (BadNet, Blended, CLA, WaNet, and IAB are insufficient.) The authors can consider attacks like SIG [1] and low frequency (Smooth) [2]. Since your method also took latent separability as an assumption, Adapt-blend and Adapt-patch attacks [3] should also be considered. Evaluating the robustness of adaptive attacks that try to evade the defense would be useful for understanding limitations.
>
> Answer: We've conducted experiments according to the reviewer's suggestions. The results are shown below:
> |                 |               | Backdoored |        | FP    |       | ANP   |       | EP    |       | CLP   |       | DRR   |      |
> |-----------------|---------------|------------|--------|-------|-------|-------|-------|-------|-------|-------|-------|-------|------|
> |                 |               | ACC        | ASR    | ACC   | ASR   | ACC   | ASR   | ACC   | ASR   | ACC   | ASR   | ACC   | ASR  |
> | ResNet-18       | AdaptiveBlend | 94.79      | 100.00 | 89.47 | 5.29  | 82.18 | 0.30  | 94.43 | 1.74  | 93.68 | 33.52 | 90.25 | 3.79 |
> |                 | SIG           | 94.01      | 98.22  | 88.94 | 45.70 | 89.38 | 2.36  | 87.36 | 30.84 | 89.75 | 94.28 | 87.10 | 0.07 |
> |                 | Smotth        | 94.59      | 100    | 87.12 | 100   | 92.65 | 81.23 | 94.24 | 3.99  | 87.24 | 89.03 | 94.03 | 3.58 |
> | WideResNet-28-1 | AdaptiveBlend | 92.37      | 100.00 | 84.77 | 51.88 | 82.06 | 42.70 | 90.45 | 5.18  | 84.40 | 74.57 | 91.13 | 0.86 |
> |                 | SIG           | 84.03      | 96.20  | 82.65 | 5.22  | 81.37 | 0.00  | 83.82 | 0.00  | 84.04 | 0.00  | 82.85 | 0.00 |
> |                 | Smotth        | 92.19      | 100    | 84.52 | 6.32  | 89.98 | 100   | 91.45 | 8.78  | 91.29 | 9.03  | 91.88 | 2.74 |
>
> Note that the latent separability mentioned in the paper [3] only considers the latent space in the penultimate layer, while our methods utilize the separability within each layer of the model. This makes our method effective even when the penultimate layer feature is inseparable.
>
>
> #### W2: The references and notations should be clarified. For example, what is the reference to Proposition 1?
>
> Answer: The reviewer's request for clarification on the references and notations is acknowledged. However, regarding the reference for Proposition 1, it should be noted that to the extent of our understanding, Proposition 1 is introduced for the first time in our manuscript. Consequently, there are no prior publications to cite for this proposition.
>
>
> #### W3: Also, the readability and organization of this paper need to be improved. It is better if an algorithm is provided.
>
> Answer: We appreciate the feedback regarding the clarity and structural aspects of our manuscript. Recognizing the value that an algorithmic representation would add, we have taken the suggestion into consideration and will include a detailed algorithm in the revised draft to enhance comprehension of our proposed method.

---

> > ### Author Response · Authors · 2023-11-21
> >
> > ### Questions:
> >
> > #### Q1: The memory and computational complexity could be analyzed more thoroughly, especially how the approach scales with larger datasets/models. Are there ways to make the optimization more efficient?
> >
> > Answer: We appreciate the inquiry regarding the computational efficiency of our approach, particularly in the context of scalability to larger datasets and models. Our method indeed incorporates strategies to enhance optimization efficiency.
> >
> > Primarily, the optimization process can be parallelized across different network layers, offering a substantial decrease in the required time for optimization. Furthermore, in scenarios with high feature dimensions or large dataset sizes, dimensionality reduction through Principal Component Analysis (PCA) can be employed prior to the directional learning phase. This step effectively reduces the computational burden. Subsequently, the learned direction within this reduced space is projected back onto the original space, thereby economizing on memory and computational demands while preserving the integrity of the optimization process.
> >
> > #### Q2: How many extension directions v_i have you used?
> >
> > Answer: Across all experiments conducted, we use only 1 direction.
> >
> > #### Q3: Modifying the weight matrix may cause a performance drop in many cases. How can your projection keep the performance?
> >
> > Answer: The reviewer has raised an important point. As we point out in our paper, rank reduction is an extension of neuron pruning (which is also a modification of the weight matrix). Established neuron pruning techniques, such as CLP and EP, have been demonstrated to maintain model performance effectively post-pruning. Our rank reduction, which 1) remove only one rank (instead of multiple ranks in neuron pruning) and 2) remove more targetedly to the direction that related to the backdoor behavior should intuitively affect less to the performance than pruning methods compared to general pruning methods. However, the exact relationship between the way we modify the weight matrices, and the performance is hard to clearly describe and can only be shown by experiments. Here we only provide an intuitive explanation.
> >
> > #### Q4: The proof needs to be more rigorous. Why use the consequence of the proof in the middle of the proof?
> >
> > Answer: We are grateful to the reviewer for highlighting a critical aspect of our proof's construction. To address this issue, we are committed to revising and strengthening our theorem to ensure its logical soundness and rigor. The revised version of our paper has included these modifications in detail. Please kindly refer to it.

---

### Official Review · Reviewer_7hoR · 2023-11-05

**Soundness:** 2 fair
**Presentation:** 3 good
**Contribution:** 3 good
**Rating:** 5
**Confidence:** 4

**Summary:**

The paper proposes a rank reduction based defense against backdoor attack. Specifically, it first gives a feature-based objective to show the optimal solution to achieve the best defense effect. He then discussed the previous defense's problem based on the given objective and proposes DRR, the rank reduction based defense where aims to find a vector that would maximize the 3rd central moments of the mixed distribution. The proposed method have been verified in CIFAR10 with several backdoor methods. The result shows the proposed method could achieve a little better performance with the state-of-art defense.

**Strengths:**

1. The paper is well-written and easy to follow with only several typos.
2. The proposed method has some good theoretical analysis and could be meaningful for the future work.

**Weaknesses:**

1. Some of theoretical analysis might be not accurate. The utility function is defined using ||R-\gamma_r (R)|| and also ||R||-||\gamma_r (R)||. However, these two value is not strict equivalent. It also happens in the definition of E(R).
2. It is unclear why the 3rd center moment would show the best performance to measure the difference. In other words, would 2nd order moment or 1st order work as well? Since 3rd order is the main metric selected, the author should explain the choice in detail.
3. The experiment is pretty insufficient. It only covers one datasets with only one poisoning rate. I suggest the author to give a more comprehensive experiments to show their proposed method's effectiveness. Some standard setting in https://github.com/SCLBD/backdoorbench is recommended.

Minor typo:
Missing \hat{x} in the definition of E(R(l).

**Questions:**

Please refer to the weaknesses part. To sum,
1. Why does ||R-\gamma_r (R)|| =||R||-||\gamma_r (R)|| along with  E(R)?
2. Why does 3rd central moment is selected?

---

> ### Author Response · Authors · 2023-11-21
>
> We appreciate the reviewer's thorough review and have taken their comments into consideration. Here are our responses to their concerns:
>
> ### Weaknesses:
>
> #### W1: Some of theoretical analysis might be not accurate. The utility function is defined using $||R-\gamma (R)||$ and also $||R||-||\gamma (R)||$. However, these two value is not strict equivalent. It also happens in the definition of $E(R)$.
>
> Answer: We thank the reviewer for pointing this out. Indeed, the equation $||R-\gamma(R)||=||R||-\gamma(R)||$ doesn't hold in the general case. However, it does hold when we use the proposed $L_{1, 1}$ norm, as defined in Definition 1. It makes sense if we specify the norm before introducing this equation. We acknowledge that the organization of this section needs to be corrected to avoid misleading the reader.
>
> #### W2: It is unclear why the 3rd center moment would show the best performance to measure the difference. In other words, would 2nd order moment or 1st order work as well? Since 3rd order is the main metric selected, the author should explain the choice in detail.
>
>
> Answer: We clarified our choice in the revision of the paper. The first central moment (the mean of the data), doesn't make sense in this context. Because the mean only affects the position of the data center, which isn't related to the direction of the mean difference. The second central moment yields similar conclusions when it is constrained with the assumptions made in the paper. However, in practice, we find the third central moment works much better. To some extent, the third central moment not only increases with the mean difference but also with the asymmetry of the two clusters. In backdoor attacks, the amount of benign data is usually much larger than poisoned data, which cannot be captured by the second central moment. That's why the third central moment performs better than the second central moment in our scenarios.
>
>
> #### W3: The experiment is pretty insufficient. It only covers one datasets with only one poisoning rate. I suggest the author to give a more comprehensive experiments to show their proposed method's effectiveness. Some standard setting in https://github.com/SCLBD/backdoorbench is recommended.
>
> Answer: Following the reviewer's suggestion, we conducted additional experiments on GTSRB. The results indicate that our method performs well in this context too. It's noteworthy that our attack with CLA on GTSRB didn't succeed, which is also the case in the BackdoorBench.
>
> |                 |              | Backdoored |        | FP    |       | ANP   |        | EP    |       | CLP   |       | DRR   |      |
> |-----------------|--------------|------------|--------|-------|-------|-------|--------|-------|-------|-------|-------|-------|------|
> |                 |              | Backdoored |        | FP    |       | ANP   |        | EP    |       | CLP   |       | DRR   |      |
> |                 |              | ACC        | ASR    | ACC   | ASR   | ACC   | ASR    | ACC   | ASR   | ACC   | ASR   | ACC   | ASR  |
> | ResNet-18       | BadNets      | 95.28      | 100.00 | 91.74 | 0.35  | 91.10 | 4.64   | 94.49 | 0.33  | 94.64 | 0.79  | 94.93 | 0.68 |
> |                 | BadNets(A2A) | 95.31      | 95.94  | 88.56 | 10.85 | 94.13 | 1.79   | 94.74 | 0.06  | 94.79 | 0.13  | 95.11 | 0.48 |
> |                 | Blended      | 95.87      | 99.88  | 90.44 | 3.00  | 91.91 | 3.20   | 94.92 | 0.29  | 94.68 | 0.97  | 95.04 | 0.79 |
> |                 | CLA          | 96.29      | 0.09   | 90.82 | 0.61  | 96.14 | 0.80   | 95.53 | 0.13  | 94.32 | 0.14  | 96.21 | 0.10 |
> |                 | Average      | 95.69      | 73.98  | 90.39 | 3.70  | 93.32 | 2.61   | 94.92 | 0.20  | 94.61 | 0.51  | 95.32 | 0.51 |
> | WideResNet-28-1 | BadNets      | 94.37      | 99.99  | 90.86 | 11.94 | 80.70 | 100.00 | 93.90 | 0.33  | 87.09 | 11.11 | 93.00 | 0.45 |
> |                 | BadNets(A2A) | 92.27      | 91.72  | 89.37 | 32.90 | 75.48 | 48.18  | 90.56 | 1.01  | 92.13 | 0.97  | 90.70 | 1.71 |
> |                 | Blended      | 94.68      | 99.75  | 90.51 | 11.35 | 86.06 | 99.52  | 92.86 | 10.26 | 93.96 | 0.26  | 92.82 | 3.07 |
> |                 | CLA          | 95.31      | 0.09   | 89.09 | 0.71  | 94.98 | 0.50   | 94.57 | 0.10  | 95.27 | 0.20  | 95.41 | 0.09 |
> |                 | Average      | 94.16      | 72.89  | 89.96 | 14.23 | 84.31 | 62.05  | 92.97 | 2.93  | 92.11 | 3.14  | 92.98 | 1.33 |

---

> > ### Author Response · Authors · 2023-11-21
> >
> > We also provide experimental results for different poisoning ratios:
> >
> > 1%:
> > |                 |              | Backdoored |       | FP    |        | ANP   |       | EP    |       | CLP   |       | DRR   |      |
> > |-----------------|--------------|------------|-------|-------|--------|-------|-------|-------|-------|-------|-------|-------|------|
> > |                 |              | ACC        | ASR   | ACC   | ASR    | ACC   | ASR   | ACC   | ASR   | ACC   | ASR   | ACC   | ASR  |
> > | ResNet-18       | BadNets      | 94.83      | 98.79 | 88.44 | 95.16  | 92.74 | 3.02  | 93.95 | 5.00  | 93.04 | 1.03  | 94.69 | 0.97 |
> > |                 | BadNets(A2A) | 94.81      | 86.52 | 89.19 | 9.97   | 91.35 | 5.32  | 94.15 | 8.90  | 93.68 | 0.82  | 94.52 | 0.66 |
> > |                 | Blended      | 95.03      | 99.99 | 89.61 | 100.00 | 93.63 | 1.68  | 91.18 | 99.73 | 93.03 | 35.66 | 94.88 | 1.84 |
> > |                 | CLA          | 94.82      | 16.18 | 89.94 | 9.84   | 92.08 | 9.26  | 94.43 | 16.80 | 91.14 | 1.22  | 94.86 | 6.34 |
> > |                 | Average      | 94.87      | 75.37 | 89.30 | 53.74  | 92.45 | 4.82  | 93.43 | 32.61 | 92.72 | 9.68  | 94.74 | 2.45 |
> > | WideResNet-28-1 | BadNets      | 92.46      | 99.92 | 86.11 | 40.66  | 73.90 | 59.50 | 90.93 | 68.04 | 90.97 | 45.56 | 90.74 | 4.41 |
> > |                 | BadNets(A2A) | 92.43      | 79.94 | 86.25 | 2.45   | 75.37 | 15.96 | 86.49 | 25.92 | 91.68 | 1.32  | 91.46 | 1.47 |
> > |                 | Blended      | 92.55      | 99.80 | 84.74 | 99.34  | 83.70 | 16.33 | 89.43 | 95.73 | 88.81 | 99.81 | 89.89 | 0.70 |
> > |                 | CLA          | 92.39      | 3.49  | 85.49 | 23.51  | 90.89 | 3.32  | 90.93 | 4.49  | 92.29 | 3.52  | 92.39 | 3.49 |
> > |                 | Average      | 92.46      | 70.79 | 85.65 | 41.49  | 80.97 | 23.78 | 89.45 | 48.55 | 90.94 | 37.55 | 91.12 | 2.52 |
> >
> > 5%:
> > |                 |              | Backdoored |        | FP    |        | ANP   |       | EP    |       | CLP   |      | DRR   |      |
> > |-----------------|--------------|------------|--------|-------|--------|-------|-------|-------|-------|-------|------|-------|------|
> > |                 |              | ACC        | ASR    | ACC   | ASR    | ACC   | ASR   | ACC   | ASR   | ACC   | ASR  | ACC   | ASR  |
> > | ResNet-18       | BadNets      | 94.14      | 99.99  | 87.31 | 100.00 | 87.92 | 1.76  | 93.60 | 2.73  | 93.05 | 1.92 | 93.90 | 1.58 |
> > |                 | BadNets(A2A) | 94.18      | 84.71  | 88.13 | 37.80  | 94.18 | 84.71 | 93.28 | 12.14 | 92.97 | 1.02 | 93.76 | 0.94 |
> > |                 | Blended      | 94.81      | 100.00 | 88.42 | 98.11  | 93.23 | 6.03  | 93.50 | 64.18 | 90.00 | 0.18 | 94.01 | 1.73 |
> > |                 | CLA          | 94.86      | 89.94  | 88.66 | 28.94  | 90.18 | 50.23 | 92.28 | 3.10  | 91.62 | 1.88 | 94.78 | 0.94 |
> > |                 | Average      | 94.50      | 93.66  | 88.13 | 66.21  | 91.38 | 35.68 | 93.17 | 20.54 | 91.91 | 1.25 | 94.11 | 1.30 |
> > | WideResNet-28-1 | BadNets      | 92.22      | 99.99  | 86.41 | 30.49  | 87.38 | 48.26 | 84.29 | 10.30 | 92.14 | 5.92 | 90.14 | 1.51 |
> > |                 | BadNets(A2A) | 92.17      | 91.26  | 84.06 | 74.53  | 84.24 | 77.23 | 91.65 | 1.53  | 91.87 | 1.45 | 92.06 | 1.70 |
> > |                 | Blended      | 91.72      | 93.76  | 84.66 | 3.23   | 83.93 | 2.23  | 89.27 | 4.39  | 89.80 | 2.06 | 89.89 | 0.70 |
> > |                 | CLA          | 92.87      | 36.23  | 85.49 | 23.51  | 86.43 | 5.09  | 85.15 | 7.87  | 90.23 | 1.90 | 91.44 | 4.63 |
> > |                 | Average      | 92.25      | 80.31  | 85.16 | 32.94  | 85.50 | 33.20 | 87.59 | 6.02  | 91.01 | 2.83 | 90.88 | 2.14 |
> >
> > and new attacks:
> > |                 |               | Backdoored |        | FP    |       | ANP   |       | EP    |       | CLP   |       | DRR   |      |
> > |-----------------|---------------|------------|--------|-------|-------|-------|-------|-------|-------|-------|-------|-------|------|
> > |                 |               | ACC        | ASR    | ACC   | ASR   | ACC   | ASR   | ACC   | ASR   | ACC   | ASR   | ACC   | ASR  |
> > | ResNet-18       | AdaptiveBlend | 94.79      | 100.00 | 89.47 | 5.29  | 82.18 | 0.30  | 94.43 | 1.74  | 93.68 | 33.52 | 90.25 | 3.79 |
> > |                 | SIG           | 94.01      | 98.22  | 88.94 | 45.70 | 89.38 | 2.36  | 87.36 | 30.84 | 89.75 | 94.28 | 87.10 | 0.07 |
> > |                 | Smotth        | 94.59      | 100    | 87.12 | 100   | 92.65 | 81.23 | 94.24 | 3.99  | 87.24 | 89.03 | 94.03 | 3.58 |
> > | WideResNet-28-1 | AdaptiveBlend | 92.37      | 100.00 | 84.77 | 51.88 | 82.06 | 42.70 | 90.45 | 5.18  | 84.40 | 74.57 | 91.13 | 0.86 |
> > |                 | SIG           | 84.03      | 96.20  | 82.65 | 5.22  | 81.37 | 0.00  | 83.82 | 0.00  | 84.04 | 0.00  | 82.85 | 0.00 |
> > |                 | Smotth        | 92.19      | 100    | 84.52 | 6.32  | 89.98 | 100   | 91.45 | 8.78  | 91.29 | 9.03  | 91.88 | 2.74 |
> >
> > The results of our experiments indicate that the performance of our proposed method exhibits robustness across a diverse scenario.

---

### Author Response · Authors · 2023-11-21

We thank all reviewers for the insightful feedback of our paper. We are glad that our **novelty** and **algorithm** are widely recognized by all reviewers. Most of the concerns are focused on **the presentation** and **the insufficient experiments** along with some **misunderstandings** of our method, which we have carefully responded in the following rebuttal. Particularly regarding the experimental section, we have significantly expanded our experimental validation, which is included in the revision of the paper.  We hope our newly added rebuttal material can address your concerns.

---

### Meta-Review · Area_Chair_ux2Z · 2023-12-05

**Metareview:**

In this paper, the authors first explored the limitations of pruning-based defense through theoretical an empirical investigations and then proposed a Directional Rank Reduction method, a so-called extended neuron pruning framework, to address the limitations.

The authors have address the comments from the reviewers.
However, after several rounds of discussions with Reviewer x3hw, he/she still is not satisfy with some responses.
In particular, the motivation or the explanation of the proposed method is questionable.
The authors should present the proposed method clearly without creating any possible misunderstnging!
Besides, Reviewer niUP raised two concerns that the authors fail to give satisfying answers regarding ``Modifying the weight matrix may cause a performance drop in many cases. How can your projection keep the performance?'' and ``The proof needs to be more rigorous. Why use the consequence of the proof in the middle of the proof?''

**Justification For Why Not Higher Score:**

However, after several rounds of discussions with Reviewer x3hw, he/she still is not satisfy with some responses.
In particular, the motivation or the explanation of the proposed method is questionable.
The authors should present the proposed method clearly without creating any possible misunderstnging!
Besides, Reviewer niUP raised two concerns that the authors fail to give satisfying answers regarding ``Modifying the weight matrix may cause a performance drop in many cases. How can your projection keep the performance?'' and ``The proof needs to be more rigorous. Why use the consequence of the proof in the middle of the proof?''

**Justification For Why Not Lower Score:**

none

---

### Decision · Program_Chairs · 2024-01-16

Reject